# Different Structures of Arabinoxylan Hydrolysates Alleviated Caco-2 Cell Barrier Damage by Regulating the TLRs/MyD88/NF-κB Pathway

**DOI:** 10.3390/foods11213535

**Published:** 2022-11-07

**Authors:** Jingwen Li, Qi Jia, Ying Liu, Daiwen Chen, Zhengfeng Fang, Yuntao Liu, Shanshan Li, Bin Hu, Caixia Wang, Hong Chen

**Affiliations:** 1College of Food Science, Sichuan Agricultural University, Yaan 625014, China; 2Institute of Animal Nutrition, Sichuan Agricultural University, Chengdu 611100, China

**Keywords:** arabinoxylan, structural features, tight junction protein, intestinal barrier, Caco-2 co-culture model, TLR signaling

## Abstract

Arabinoxylan (AX) has been associated with alleviating intestinal barrier damage, and different structures of AX give rise to different effects on the intestinal barrier. This study investigated the main structural characteristics of AX, whose functional properties are attributed to alleviating intestinal barrier damage, and clarified their underlying mechanisms. An in vitro Caco-2 cell model was established to investigate the intestinal barrier effects of AX with various degrees of substitution (Ds) and molecular weight (Mw), with an added MyD88 inhibitor to verify the signaling pathways. Arabinoxylan treated with endo-1,4-*β*-xylanase (AX_X_) with higher Ds and Mw showed stronger physiological activity, which might be correlated with the uronic acid and bound ferulic acid contents in AX_X_. Moreover, AX_X_ alleviated the intestinal barrier damage by upregulating the transepithelial electrical resistance (TER) and alleviating the decrease of claudin-1 (*p* < 0.05). AX_X_ regulated the expression of inflammatory factors IL-2, TNF-α, IL-6 and IL-10 (*p* < 0.05). In addition, AX_X_ reduced the intestinal barrier damage induced via inhibiting the TLRs/MyD88/NF-κB pathway and activating the TLRs/PKC pathway. Thus, AX with higher Ds and Mw might be better in alleviating intestinal barrier damage, and MyD88 might be the key point of AX_X_ to identify these signaling pathways.

## 1. Introduction

The integrity of epithelial cell and tight junction (TJ) proteins is often referred to as the “intestinal barrier” [1,2]. The intestinal epithelial barrier plays a key role in preventing the transfer of pathogenic microorganisms and toxic substances from the intestinal lumen to the systemic circulation [3]. The loss of barrier integrity can lead to the invasion of harmful substances in the intestinal lumen, which can cause different diseases such as inflammatory bowel disease and extraintestinal autoimmune diseases [4,5]. TJ connectivity plays an important role in maintaining the structural and functional integrity of the intestinal barrier by regulating paracellular transport. In recent studies, increased intestinal permeability due to decreased TJ-related protein (such as zonula occludens family proteins, occludin and claudins) expression and epithelial barrier function was observed [6].

Toll-like receptors (TLRs) are a class of transmembrane proteins that trigger cell signal transduction. Toll-like receptor 2 (TLR2) is a potential therapeutic target for gastrointestinal diseases, in which TJ-related intestinal epithelial barrier disruption might be the main feature [7,8]. Toll-like receptor 4 (TLR4) is the best characterization protein of pathogen recognition receptors and plays a homeostasis role in maintaining TJ protein expression and affecting intestinal permeability [9]. Myeloid differentiation factor 88 (MyD88) is the core of transduction of extracellular stimulation via TLRs. After specific recognition, intracellular signal transduction mainly depends on MyD88-dependent and MyD88-independent pathways. Nuclear factor-κB (NF-κB) is an MyD88-dependent node of TLR’s downstream inflammatory signaling pathway, and the activation of NF-κB and synthesis of inflammatory cytokines not only intensify the inflammatory response but also compromise intestinal integrity [10]. Protein kinase C (PKC) is a family of serine- and threonine-specific protein kinases that mediates numerous cellular processes in a tissue-specific manner. Studies have shown that the activation of PKC enhances the integrity of TJ, and the PKC pathway might enhance the intestinal TJ barrier [11].

Arabinoxylan (AX) is a complex polymer that is composed of a backbone of *β*-d-xylopyranose residues connected by *β-*(1,4)-glycosidic bonds, which is substituted by arabinose residues at the C(O)-2 and/or C(O)-3 position [12]. In addition, arabinose can be attached with some phenolic acids. For example, ferulic acid (FA) can be covalently linked to the C5 position of the α-L-arabinofuranosidase residue via an ester bond [13]. Many in vivo studies have shown that AX has a protective effect on the intestinal barrier [14,15,16]; however, its specific mechanisms in vitro remain unclear. Studies have shown that the biological activity of AX is highly dependent on the chemical structure of AX [17]. In vitro, Mendis, et al. [18] found that increased arabinose substitution in AX might be better at reducing inflammation in colon cancer cells. AX with low-branched hydroxycinnamate and higher DS exhibit stronger antioxidant activity [19]. Our previous study showed that triticale bran AX with higher molecular weight had higher antioxidant levels [12]. These studies showed that the Mw or degree of arabinose substitution is closely related to the physiological function of AX. Previous studies tended to focus on the effect of a single factor (Mw or Ds) on the physiological function of AX. However, little information is available regarding the combined effect of these two factors on physiological function, and further verification is needed. Therefore, in the present study, four arabinoxylan hydrolysates (AXHs) with different degrees of substitution and molecular weight were prepared for cell culture to investigate the relationship between different unique structures of AX and the effects of alleviating Caco-2 cell permeability. Furthermore, the potential pathways of action were explored.

## 2. Materials and Methods

### 2.1. Materials and Reagents

The bran was acquired from Xinxiang Agricultural Development Company Limited (Xinxinag China). Neutral protease (activity: 100 μ/mg protein) was acquired from Shanghai Yuanye Biotechnology Company Limited (Shanghai, China). α-amylase from *Aspergillus oryzae* and endo-1,4-*β*-xylanase from *Penicillium* were acquired from Shanghai Ryon Biological Technology Company Limited (Shanghai, China). *α*-L-arabinofuranosidase B21 from *Bacteroides ovatus* was acquired from Megazyme International Ireland (Wicklow, Ireland). Solvents and reagents were of analytical grade throughout the experiment.

### 2.2. Preparation of Arabinoxylan Hydrolysates

AX was dispersed in citric acid solutions with the content of 0.1 M and 0.15 M [20,21]. After a two hour 95 °C water bath, NaOH was added. The mixture was centrifuged, precipitated with ethanol, redissolved and lyophilized to yield the hydrolysate AX_C1_ and AX_C_. AX_C_ was added with 0.15 M HCl to obtain hydrolysate AX_H2_. *α*-L-arabinofuranosidase was added in AX solution and oscillated in a 40 °C water bath for 24 h. After centrifugation, samples were freeze-dried to obtain hydrolysate AX_AE_. Endo-1,4-*β*-xylanase was added in the AX suspension. After centrifugation, collected and added anhydrous ethanol. After centrifugation for 15 min, the precipitation was collected and redissolved. The supernatant was extracted after repeated centrifugation for 15 min and freeze-dried to obtain hydrolysate AX_X_.

### 2.3. Structural Characterisations of Arabinoxylan Hydrolysates

#### 2.3.1. Molecular Weight Measurement

Samples were dissolved and the sample solution was filtrated [17]. A Refractive index detector (Shimadzu RID-20, Shimadzu, Kyoto, Japan) and a gel filtration column (TSK GMPWXL, 7.8 mm × 300 mm, TOSOH, Yamaguchi, Japan) were used for analysis. A quantity of 0.1 N NaNO_3_ and 0.06% NaN_3_ aqueous solution as mobile phase and flow rate of 0.6 mL/min. Standard dextrans were used to obtain the standard curve.

#### 2.3.2. Monosaccharide Composition Measurement

Determination with the method illustrated by Yuan, et al. [22]. A Dionex UltiMate 3000 HPLC system (Thermo Fisher Scientific, Waltham, MA, USA) was used to analyze 1-Phenyl-3-methyl-5-pyrazolone (PMP) derivatives. A quantity of 20 μL of PMP derivatives were injected into the HPLC system.

#### 2.3.3. ^1^H NMR Analysis

The analysis was conducted in accordance with a previous article [23]. Here, after exchange with D_2_O, the samples were finally dissolved in pure D_2_O. A 400.00 MHz spectrometer (JNM-ECZ400S, JEOL, Tokyo, Japan) was used to obtain the ^1^H spectra of samples. The ^1^H chemical shifts (ppm) were referenced to a D_2_O signal at 4.790 ppm at 25 °C.

#### 2.3.4. FT-IR Analysis

The analysis was conducted in accordance with a previous article [22]. Fourier transform infrared spectra of freeze-dried samples were recorded at room temperature with an FTIR spectrometer (Thermo Fisher Scientific, Waltham, MA, USA).

#### 2.3.5. Ferulic Acid Determination

FA content was measured by spectrophotometry [24]. The sample was dissolved in 1 mL of ultrapure water. A quantity of 900 μL of glycine-NaOH buffer (pH 10, 0.04 M) was added to 100 μL of the sample solution. The absorbance values were at 345 and 375 nm.

### 2.4. Establishment of the Cell Model

#### 2.4.1. Cell Culture

Caco-2 cell culture was performed based on Fang, et al. [25]. Human colon cells (Caco-2) were selected as the research object. The Caco-2 cell line (ATCC, Manassas, VA, USA), derived from human colon cancer cells, in DMEM containing 10% fetal bovine serum, 50 U/L penicillin-streptomycin and 1% non-essential amino acid at 37 °C, 5% CO_2_ high-sugar medium, was cultured in a 25 cm^2^ cell culture flask, replacing the culture medium every 1–2 days. After 5–7 days, the cells had grown to confluence and passaged at 1:2 or 1:3.

#### 2.4.2. Treatment of Cells with Arabinoxylan Hydrolysates

The treatment was performed based on Wu, et al. [26]. The cells were starved for 12 h before the experiment on serum-free media, and the treatment groups were added with AXHs (400 μg/mL) and different concentrations of lipopolysaccharide (LPS, μg/mL) after 1 h preincubation. The control group (without LPS and AXH), negative control group (with LPS) and treatment groups (with LPS and AXH) were set. The resistance values of each group were detected after 24 h of culture, and the experiment was repeated three times.

#### 2.4.3. Treatment of Cells with MyD88 Inhibitor

The treatment was performed based on Song, et al. [27]. The cells were starved for 12 h before the experiment (adding serum-free medium), and all groups were added with MyD88 inhibitor. After 1 h, the AXHs with the best screening effect was added (μg/mL). After 2 h, LPS (μg/mL) was added. The control group (no LPS and no AXH), negative control group (only LPS), treatment group I (LPS and AXH, no MyD88 inhibitor) and treatment group II (LPS, the best effect AXH and MyD88 inhibitor) were set. The resistance values of each group were detected after 24 h of culture, and the experiment was repeated three times.

### 2.5. Determination of Epithelial Monolayer Resistance

Determination was conducted with the method illustrated by Wu, et al. [28]. The cells were starved for 12 h before the experiment (adding serum-free medium), pre-incubated for 1 h and then added with LPS (μg/mL). Solvent was used as the control group, and the resistance value of each group was detected after 24 h of culture. The short and long arms of the cell resistor electrode were inserted into the upper and lower chambers of the transwell chamber, respectively. Three different points of the chambers were measured, and the average value of the measured values was taken.

### 2.6. Assessment of Protein Expression by Western Blot

Assessment by Western blot was performed according to Wu, et al. [29]. The Caco-2 cell culture medium was aspirated and rinsed with PBS twice. The cells were digested by trypsin, and the pipetted cells were transferred into a 1.5 mL EP tube. A total of 150 μL of RIPA lysate containing PMSF was added to the centrifuge tube and centrifuged at 4 °C, and the supernatant after centrifugation was prepared with 10% polyacrylamide gel to prepare SDS-PAGE gel. About 50 μg of protein sample was prepared and loaded. Firstly, electrophoresis was performed at a constant voltage of 80 V and then 100 V. The protein in the gel was transferred to the PVDF membrane by transfer membrane and blocked with TBST buffer containing 5% skimmed milk powder. Then, the primary antibody was incubated overnight at 4 °C. The secondary antibody was incubated at room temperature for 1 h. ImageJ image processing software was used to analyze the Mw and net optical density value of the target band. The average of five experimental results was regarded as the relative protein content. β-actin was the internal reference gene.

### 2.7. Quantification of Gene Expression Using Real-Time PCR

The cells were collected and washed with PBS buffer three times. Then, 1 mL of Trizol reagent was added to each well to extract the total RNA of the cells. RNA concentration and purity were detected and reversed transcription was synthesized into cDNA. The target gene sequence was acquired from the NCBI biological information website, and Primer 5 primer design software was used to construct the gene primers. A 10 μL SYBR Green reaction system was used for RT-PCR detection, β-actin and GADPH were used as internal reference genes for calculation, and the target gene level was expressed in terms of relative expression. The calculation method of the results was reported in the study of Pfaffl, et al. [30].

### 2.8. Enzyme-Linked Immunosorbent Assay

The culture medium was discarded. The cells were collected and washed three times with PBS buffer. An ELISA kit was used to determine the cytokines IL-2, IL-6, IL-10 and TNF-α in the Caco-2 cell culture medium content. The determination method was carried out in accordance with the ELISA kit’s instructions.

### 2.9. Statistical Analysis

All data were entered into Excel 2019. OriginPro 9.1 was used for mapping. IBM SPSS Statistics 26 mathematical software was used for statistical analysis of experimental data, and analysis of variance (ANOVA) was used to verify the statistical differences between groups. The difference was statistically significant with *p* < 0.05, and the results were expressed as mean ± standard deviation.

## 3. Results

### 3.1. Structural Characterisations of Arabinoxylan Hydrolysates

#### 3.1.1. Molecular Weight and Degree of Substitution of Arabinoxylan Hydrolysates

The Mw and Ds of AXHs are shown in Table 1. The monosaccharide composition of AXH was mainly composed of xylose and arabinose, following with glucose and galactose. Different AXHs had a different ratio of arabinose to xylose and AX_X_ and AX_AE_ contained more arabinose. Furthermore, the Mw of AXH acquired by endo-1,4-*β*-xylanase treatment was higher than that acquired by *α*-L-arabinofuranosidase treatment. The Mw is mainly distributed between 2.67 × 10^3^ Da to 6.43 × 10^5^ Da. Furthermore, in the acid-treated AX_C1_ and AX_H2_, the Mw of AX_H2_ was lower than that of AX_C1_.

#### 3.1.2. ^1^H NMR Analysis of Arabinoxylan Hydrolysates

The structural change of AXH was verified by ^1^H NMR. The NMR spectra of all samples showed 3–6 main peaks in the chemical shift range of 5.15–5.40 ppm. The signal peak of the arabinose residue was connected to the C(O)-3 position of the xylose residue on the main chain at a chemical shift of 5.38 ppm. The signal peak around 5.27 and/or 5.21 ppm corresponding to xylose residues was monosubstituted by arabinose residues at the C(O)-2 and/or C(O)-3 position. The chemical shift of xylose residues was monosubstituted at the C(O)-3 and/or C(O)-2 position at 4.50 and/or 4.60 ppm. The chemical shift of xylose residues was disubstituted at 4.62 ppm. As shown in Figure 1, the contents of xylose residues in AX_H2_ and AX_AE_ were higher than those in AX_X_ and AX_C1_.

#### 3.1.3. FT-IR Spectrum Analysis of Arabinoxylan Hydrolysates

As shown in Figure 2, a broad and strong peak appeared near 3410 cm^−1^. The range of 3000–2800 cm^−1^ were produced by C-H vibration. The band between 1700–1600 cm^−1^ was the C=O asymmetric stretching vibration in COO-. The absorption band between 1400–1300 cm^−1^ was the C=O symmetric stretching vibration in COO-. Frequency band signals between 1200 cm^−1^ and 800 cm^−1^ were associated with a specific polysaccharide structure composed of pyranose ring vibration, which overlapped with C-C and C-OH stretching vibration and glycosidic bond (C-O-C) vibration. AX_AE_ had a relatively strong absorption peak at 1074.50 cm^−1^. These were the typical structural features of AX.

#### 3.1.4. Ferulic Acid Content Analysis

FA was a major bound phenolic acid linked to AX mainly with ester bounds, which affected its physical and chemical properties. Free FA content was higher than that of bound FA (Figure 3). Bound FA content in AX_X_ was higher than that acquired by acid and *α*-L-arabinofuranosidase treatment (*p* < 0.05).

### 3.2. Regulating Effects of AXH on the Intestinal Barrier in the Caco-2 Model

#### 3.2.1. Transepithelial Electrical Resistance Measurement

About 1 μg/mL of LPS was used as the optimal LPS emergency concentration for our study. The alleviating effect of each sample on LPS-induced intestinal epithelial barrier integrity destruction was monitored by transepithelial electrical resistivity (Figure 4). Our research showed that after treatment with the LPS group, the resistivity of Caco-2 intestinal epithelial cells decreased by 9%. The TER value of the LPS + AX_X_ group was higher than the LPS + AX_C1_ group (407.12 ± 34.08 Ω/cm^2^), and there was no significant difference among other groups.

#### 3.2.2. Tight Junction Protein Expression in Caco-2 Cells

Claudin-1, occludin and zonula occludens (ZO)-1 were down-regulated under LPS conditions (*p* < 0.05), and this effect was changed by the different structures of AXHs (Figure 5A). Compared with LPS, AX_X_ increased the expression of claudin-1 (*p* < 0.05), and there was no significant expression of occludin and ZO-1 in the AX_X_ group (*p* > 0.05). As shown in Figure 5B, the expression of claudin-1 was down-regulated in the LPS + AX_X_ + MyD88 inhibitor group (*p* < 0.05), whereas there was no significant change in occludin and ZO-1 (*p* > 0.05).

#### 3.2.3. Signaling Pathway in Caco-2 Cells

As shown in Figure 6A,B, compared with the LPS group, the AX_X_ and AX_H2_ groups up-regulated the expression of TLR2 and TLR4 (*p* < 0.05). After MyD88 inhibitor was added, the expression of TLR2 and TLR4 was down-regulated compared with the LPS group (*p* < 0.05). The LPS + AX_X_ group up-regulated the expression of MyD88 compared with the LPS group, whereas this change decreased in the MyD88 inhibitor group (*p* < 0.05) (Figure 6C). Figure 6D,E shows that LPS-induced PKC and NF-κB expression in Caco-2 cells was upregulated and inhibited, respectively, compared with that in the AX_X_ group (*p* < 0.05). In the LPS + AX_X_ + MyD88 inhibitor group, NF-κB expression was up-regulated (*p* < 0.05), whereas PKC expression was not significantly changed compared with that in the AX_X_ group (*p* > 0.05) (Figure 6D,E).

#### 3.2.4. Inflammatory Cytokines

As shown in Figure 7A–D, the LPS + AX_X_ group down-regulated TNF-α, IL-6 and IL-10 expression and up-regulated IL-2 expression compared with the LPS group (*p* < 0.05). In addition, compared with the LPS + AX_X_ group, the addition of MyD88 inhibitor up-regulated TNF-α, IL-6 and IL-10 expression and down-regulated IL-2 expression (*p* < 0.05) (Figure 7A–D).

## 4. Discussion

Our previous studies marked the effect of different dietary fibers on improving the intestinal barrier, and found that wheat bran fiber had the best effect [31]. Further studies found that the intestinal epithelial barrier function improvement exerted by wheat bran was related to its AX content [32]. AX with different structural characteristics exhibited different biological activities, and there was a strong correlation between the molecular structure and physiological functions of AX. In our previous study, AXHs with different structures were prepared by different methods [21]. According to their substitution degree and molecular weight difference, four representative AXHs were identified, including AX_X_ with high Ds and high Mw, AX_AE_ with high Ds and low Mw, AX_C1_ with low Ds and high Mw and AX_H2_ with low Ds and low Mw. The arabinose residue of AX performed different structures along the xylose backbone [23]. The ^1^H NMR results showed that AXHs except AX_AE_ had disubstituted arabinose residues, which indicated the role of *α*-L-arabinofuranosidase, and the higher A/X ratio of AX_AE_ might be related to the region of monosubstituted arabinose residue and disubstituted xylose residues. In addition, AXHs except AX_X_ had mono- and/or disubstituted xylose residues, indicating that endo-1,4-*β*-xylanase selectively removes xylose residues, and the higher degree of AX_X_ could be confirmed (Ara/Xyl ratio: 1.17). Although the A/X ratio of AX_AE_ and AX_X_ were similar, the substitution patterns of these polymers were not the same. The A/X ratio of AX_C1_ was higher than AX_H2_ due to the former monosubstituted xylose residue. FT-IR spectrum results showed that the high Ds of arabinose at xylose residue C3 was indicated by the low-intensity peaks at 988.18 and 1154.83 cm^−1^, which proved that AX_AE_ had more branched structures and was consistent with the high Ds of AX_AE_. A strong absorption peak of AX_X_ at 1650 cm^−1^ was associated with water in the sample, which might be related to the fact that AX_X_ had a highly mono- and/or disubstituted arabinose. The solubility of AX was reported to be determined by its structure, with a higher A/X ratio promoting its solubility [17]. Moreover, the solubilization of arabinose substituents might be due to the prevention of intermolecular aggregation of unsubstituted xylose residues.

Uronic acid was observed in AX_X_ and AX_C1_. Studies had shown that uronic acid polysaccharides activated MAPK and NF-κB signaling pathways by inducing the release of inflammatory cytokines. Therefore, AX with uronic acid might have more physiological properties [33,34]. Hromádková, et al. [35] revealed that the aggregation formed through the crosslinking of FA substituents might be responsible for the high Mw of AX, which was verified by the higher proportion of FA binding of AX_X_ in our work. FA played an important role in the physiological functions of AX, especially in the antioxidant capacity of AX, but it contributed less to the effect of AX on the Caco-2 cell barrier. Zhang, et al. [16] suggested that FA bound to arabinose in AX to regulate intestinal barrier damage through gut microbiota. We speculated that AX_X_ was more biologically active in the intestinal epithelial cell due to its unique structure. The molecular mechanisms behind this effect remains to be elucidated.

A decrease in TER was generally considered a reference index for cell damage or death, whereas the TER values might be restored and increased by LPS co-culture with AX_X_, which might be better for improving the permeability of Caco-2 cells. TJs surrounded and anchored adjacent cells. The depletion of TJ protein expression impaired intestinal barrier permeability and the ability to maintain TER [36]. Claudin-1, occludin and ZO-1 were three important TJ proteins. Claudin-1 interacted with other connexins in neighboring cells and formed fences that regulated the permeability of TJ complexes. Occludin could bind with adjacent cells through the outer part of the cell membrane, which conduced to control the permeability between cells. ZO-1 was mainly located at the boundary of adjacent epithelial cells, which anchored occludin and claudin-1 to the cytoskeleton and co-controlled intestinal barrier permeability [37,38]. In this study, LPS reduced the expression of ZO-1, occludin and claudin-1. Claudin was offset by AX_X_. Claudin has been reported to largely determine the paracellular ion permeability at TJs [39]. We suggested that AX_X_ maintained the integrity of the intestinal epithelial barrier by promoting the expression of claudin-1.

Pro-inflammatory cytokines (IL-1, IL-2, IL-6, etc.) had been proven to be interfering factors of the intestinal barrier. They lead to the further deterioration of intestinal epithelial function by inhibiting TJ protein expression [40]. In this study, AX_X_ induced the expression of interleukin-2 (IL-2), inhibited the expression of interleukin-6 (IL-6), interleukin-10 (IL-10) and tumor necrosis factor-α (TNF-α) to enhance the intestinal immune function of the host, which was consistent with the study of Mendis, et al. [18]. These results suggested that AX_X_ production played an important role in improving intestinal barrier and permeability by regulating TJ protein expression and inflammatory cytokine secretion.

Next, we tried to reveal the mechanism of AX_X_ in alleviating the intestinal barrier in Caco-2 cells. Our previous study found that the up-regulation of TJ protein gene expression in intestinal epithelial cells by wheat bran dietary fiber was accompanied by the up-regulation of TLR2 expression [31]. Dietary fiber inulin-type fructans could modulate TLR2 to enhance the intestinal barrier and prevented pathogens from entering the host [41]. TLR4-mediated intestinal barrier dysfunction was considered to be a key factor in the initiation and enhancement of gastrointestinal injury [42]. In this study, the expression of TLR2 and TLR4 was up-regulated in the AX_X_ and AX_H2_ groups, which suggested that structurally different AX had similarities in regulating intestinal barrier damage. AX_X_ and AX_H2_ might induce cytokine production in intestinal epithelial cells in a TLR2- and/or TLR4-dependent manner. The similarity of AXH biologically affected with different Mw and Ds might be due to the similarity of glycosidic bonds and changes in the structure and number of arabinose substitutions in the xylose backbone [43]. In addition, we found that the infrared spectra of AX_H2_ and AX_X_ were similar. However, according to the data of this study, AX_X_ with high Ds might exert steric hindrance on the formation of intermolecular crosslinks and thus be easily dispersed into the reactive mixture to obtain a better performance.

MyD88 was an important signaling pathway regulating LPS-induced intestinal epithelial TJ permeability [44]. It had been widely reported that polysaccharides exerted their regulatory role by acting on MyD88. Based on these studies, a model with additional MyD88 inhibitor in the LPS + AX_X_ group was proposed to explore and verify the signaling pathways by which AX_X_ alleviated LPS-induced changes in barrier permeability. Our results confirmed that the expression of claudin-1 was affected by the addition of MyD88 inhibitor. The altered expression of inflammatory factors after MyD88 inhibitor suggested that this response might be related to MyD88-dependent TLR signaling. NF-κB activation might be an important pathway of TJ barrier dysfunction induced by pro-inflammatory cytokines. Supplementation with AX_X_ reduced the damage of intestinal mucosa primarily by inhibiting the MyD88-dependent NF-κB pathway and promoting TJ protein expression, which was demonstrated with the addition of MyD88 inhibitor. The phosphorylation of downstream PKC was associated with the activation of TLR2 [45]. Studies had shown that PKC activation enhanced the integrity of TJs, and the PKC pathway might enhance the intestinal TJ barrier [40]. PKC expression was not significantly changed in the LPS + AX_X_ + MyD88 inhibitor group, which suggested that the function of PKC in a pathway was not dependent on MyD88. Given that the addition of AX_H2_ had no effect on MyD88 expression, we hypothesized that AX_X_ alleviated intestinal barrier damage by stimulating the TLRs/PKC and inhibiting the TLRs/MyD88/NF-κB signaling pathways.

## 5. Conclusions

The results of this study demonstrated that the function of AXH to alleviate Caco-2 cell permeability was related to its fine structure and physicochemical properties. AX_X_ with higher Mw, Ds and bound FA levels was more beneficial to the improvement of barrier function. Furthermore, AX_X_ promoted epithelial barrier integrity by influencing TER, claudin-1 and inflammatory cytokines via the TLRs/MyD88/NF-κB and TLRs/PKC signaling pathways, which was verified by additional MyD88 inhibitor. Whether TLR2 and TLR4 are involved in the two signaling pathways remains unclear. As far as the current results are concerned, further research is required to obtain AX with detailed structure and characteristics, which will lay a solid foundation on the precise regulation of AX.

## Figures and Tables

**Figure 1 foods-11-03535-f001:**
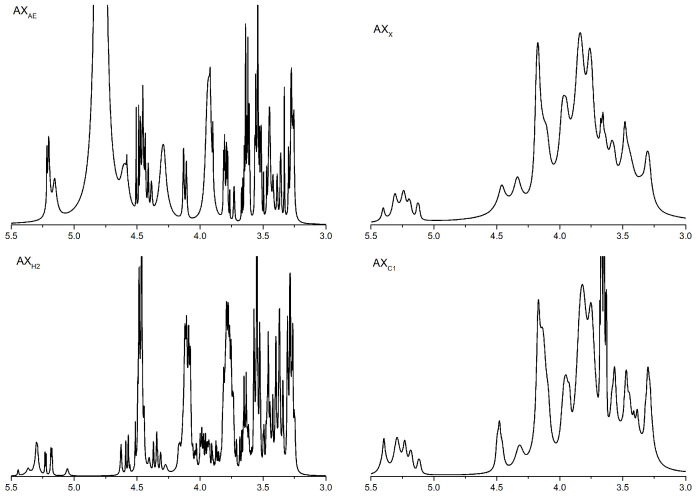
^1^H NMR analysis of AXH. AX_AE_, triticale bran arabinoxylan was treated with *α*-L-arabinofuranosidase; AX_C1_, triticale bran arabinoxylan was treated with 0.1 M citric acid; AX_H2_, triticale bran arabinoxylan was treated with 0.15 M HCL; AX_X_, triticale bran arabinoxylan was treated with endo-1,4-*β*-xylanase.

**Figure 2 foods-11-03535-f002:**
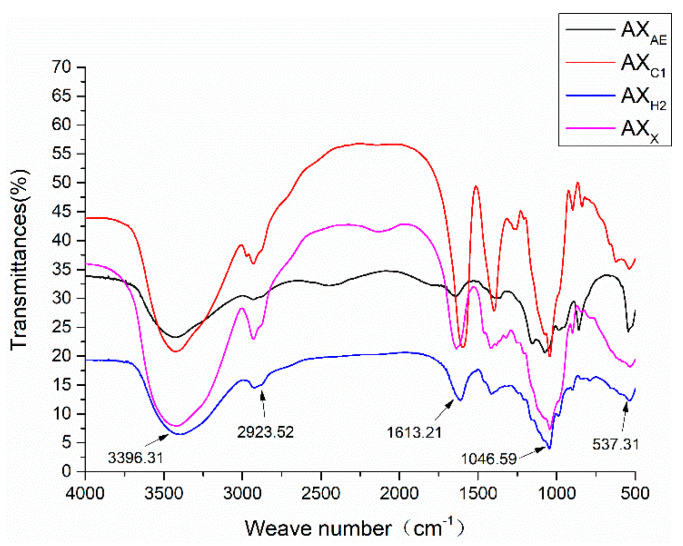
FT-IR spectra of AXH. AX_AE_, triticale bran arabinoxylan was treated with *α*-L-arabinofuranosidase; AX_C1_, triticale bran arabinoxylan was treated with 0.1 M citric acid; AX_H2_, triticale bran arabinoxylan was treated with 0.15 M HCL; AX_X_, triticale bran arabinoxylan was treated with endo-1,4-*β*-xylanase.

**Figure 3 foods-11-03535-f003:**
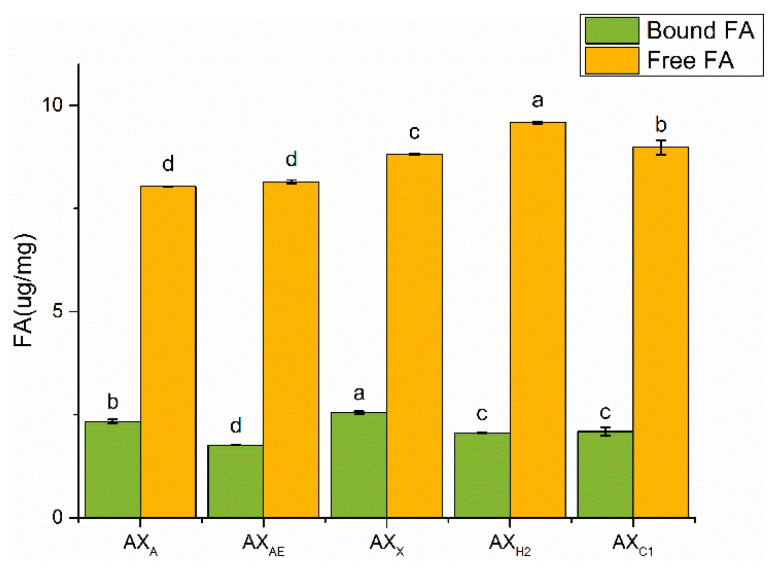
The content of FA in AXH. AX_AE_, triticale bran arabinoxylan was treated with *α*-L-arabinofuranosidase; AX_C1_, triticale bran arabinoxylan was treated with 0.1 M citric acid; AX_H2_, triticale bran arabinoxylan was treated with 0.15 M HCL; AX_X_, triticale bran arabinoxylan was treated with endo-1,4-*β*-xylanase. Values represented in this figure consist of mean ± standard deviation (*n* = 3). Values with different superscript letter are significantly different (*p* < 0.05).

**Figure 4 foods-11-03535-f004:**
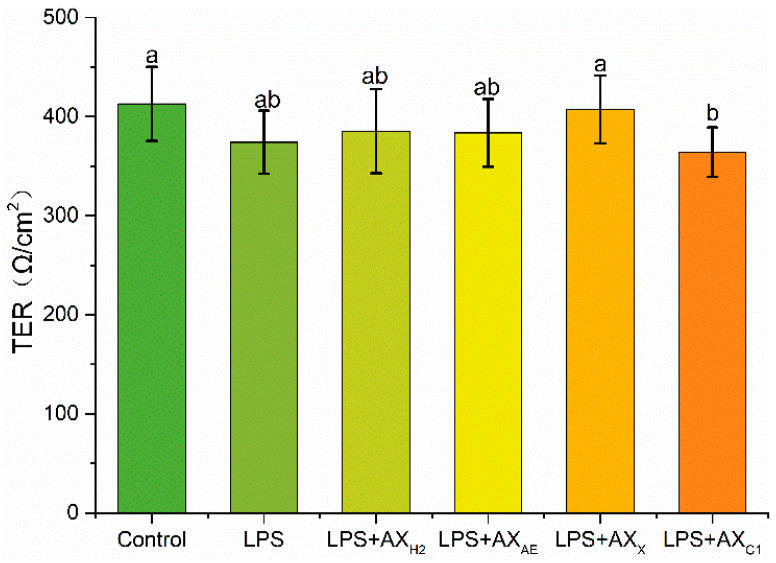
Transepithelial electrical resistance after treated with LPS. Control, without LPS or arabinoxylan hydrolysates; AX_H2_, triticale bran arabinoxylan was treated with 0.15 M HCL; AX_AE_, triticale bran arabinoxylan was treated with *α*-L-arabinofuranosidase; AX_C1_, triticale bran arabinoxylan was treated with 0.1 M citric acid; AX_X_, triticale bran arabinoxylan was treated with endo-1,4-*β*-xylanase. Values represented in this figure consist of mean ± standard deviation (*n* = 3). Values with different superscript letter are significantly different (*p* < 0.05).

**Figure 5 foods-11-03535-f005:**
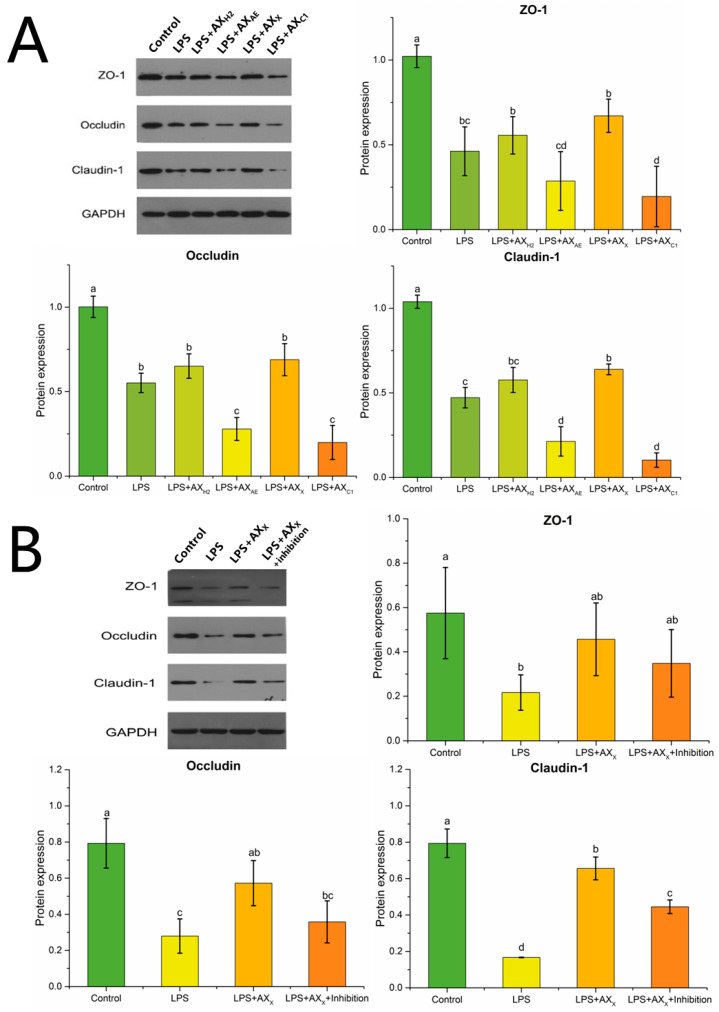
Tight junction protein expression in Caco-2 cells treated with (**A**) AXHs and (**B**) MyD88 inhibitor. Control, without LPS or arabinoxylan hydrolysates; AX_H2_, triticale bran arabinoxylan was treated with 0.15 M HCl; AX_AE_, triticale bran arabinoxylan was treated with *α*-L-arabinofuranosidase; AX_C1_, triticale bran arabinoxylan was treated with 0.1 M citric acid; AX_X_, triticale bran arabinoxylan was treated with endo-1,4-*β*-xylanase; Inhibitor, MyD88 inhibitor. Values represented in this figure consist of mean ± standard deviation (*n* = 3). Values with different superscript letter are significantly different (*p* < 0.05).

**Figure 6 foods-11-03535-f006:**
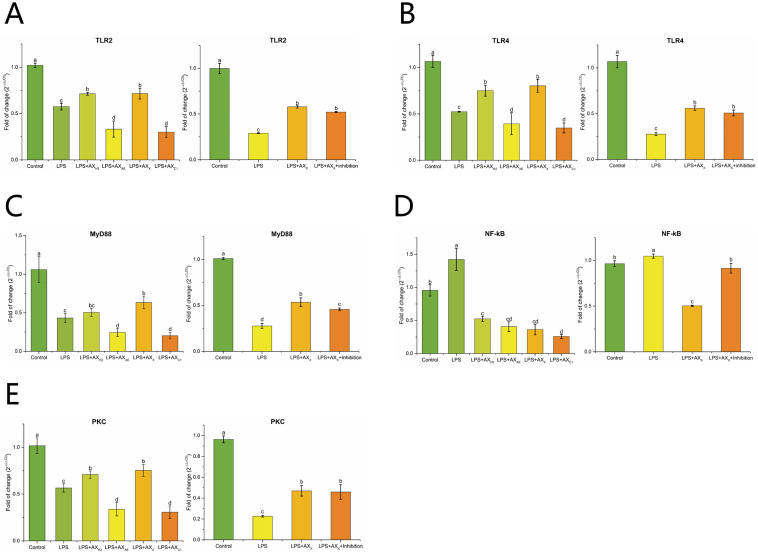
The effect of AXHs and MyD88 inhibitor on the expression of TLR2 (**A**), TLR4 (**B**), MyD88 (**C**), NF-κB (**D**), PKC (**E**) in LPS-induced Caco-2 cells. Control, without LPS or arabinoxylan hydrolysates; AX_H2_, triticale bran arabinoxylan was treated with 0.15 M HCl; AX_AE_, triticale bran arabinoxylan was treated with *α*-L-arabinofuranosidase; AX_C1_, triticale bran arabinoxylan was treated with 0.1 M citric acid; AX_X_, triticale bran arabinoxylan was treated with endo-1,4-*β*-xylanase; Inhibitor, MyD88 inhibitor. Values represented in this figure consist of mean ± standard deviation (*n* = 3). Values with different superscript letter are significantly different (*p* < 0.05).

**Figure 7 foods-11-03535-f007:**
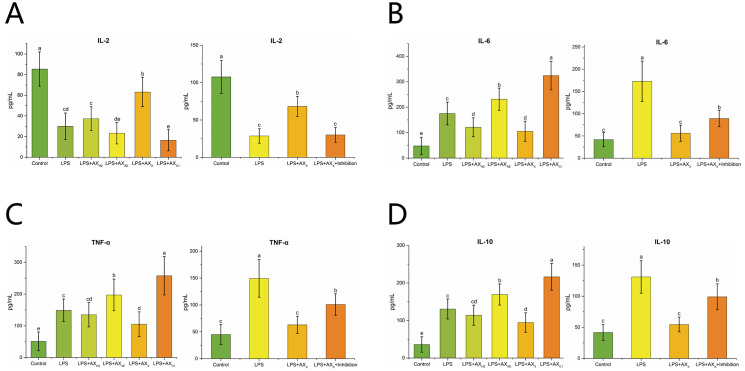
The effect of AXHs and MyD88 inhibitor on the expression of IL-2 (**A**), IL-6 (**B**), TNF-α (**C**), IL-10 (**D**) in LPS-induced Caco-2 cells Control, without LPS or arabinoxylan hydrolysates; AX_AE_, triticale bran arabinoxylan was treated with *α*-L-arabinofuranosidase; AX_X_, triticale bran arabinoxylan was treated with endo-1,4-*β*-xylanase; AX_H2_, triticale bran arabinoxylan was treated with 0.15 M HCl; AX_C1_, triticale bran arabinoxylan was treated with 0.1 M citric acid; Inhibitor, MyD88 inhibitor. Values represented in this figure consist of mean ± standard deviation (*n* = 3). Values with different superscript letter are significantly different (*p* < 0.05).

**Table 1 foods-11-03535-t001:** Molecular weight and monosaccharide composition of AXHs.

Samples	Monosaccharide Composition (Molar Ratio)				A/X	Average Molecular Weight (Da)	Polydispersity Index (Mw/Mn)
	Xylose Arabinose Glucose Galactose						
AX_H2_	1.00	0.15	0.04	0.05	0.15	7.47 × 10^3^	1.81
AX_AE_	1.00	0.97	0.02	0.05	0.97	2.67 × 10^3^	1.24
AX_X_	1.00	1.17	0.02	0.07	1.17	6.43 × 10^5^	2.28
AX_C1_	1.00	0.83	0.04	0.05	0.83	2.36 × 10^5^	2.27

AX_H2_, triticale bran arabinoxylan was treated with 0.15 M HCL; AX_C1_, triticale bran arabinoxylan was treated with 0.1 M citric acid; AX_AE_, triticale bran arabinoxylan was treated with *α*-L-arabinofuranosidase; AX_X_, triticale bran arabinoxylan was treated with endo-1,4-*β*-xylanase.

## Data Availability

Data is contained within the article.

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
