# Peer review of "Different Structures of Arabinoxylan Hydrolysates Alleviated Caco-2 Cell Barrier Damage by Regulating the TLRs/MyD88/NF-κB Pathway"

_foods, 2022, doi:10.3390/foods11213535_

Round 1

Reviewer 1 Report

 The manuscript entitled '' Different structures of arabinType of the Paperoxylan hydroly sates alleviated Caco-2 cell barrier damage by regulating the TLRs/MyD88/NF-κB pathway'' is interesting and organized. However, the authors need to make some changes.

-        In the title, some words are capitalized, and some are small. Please unify them.

-        Please, check the abbreviations throughout the manuscript. You use AX, AXs, and sometimes AXx.

-         β symbol should be italic in β-d-xylopyranose, β-(1,4)-glycosidic, and other words.

-        Line 108  1H NMR instead of H NMR.

-        The author should give the reference in the Materials & Methods section, Treatment of cells with arabinoxylan hydrolysates and  Treatment of cells with MyD88 inhibitor.

-        1H NMR results need more clarification.

-        Please, write Figure instead of Fig.

-        The authors should highlight their study. More discussion could be put in this part, and some comparison between this study and other investigations should be mentioned.

-        Some grammatical, alignment and typographical errors are noted in the manuscript, and they should be thoroughly checked and corrected throughout the manuscript.

-        Some symbols or letters on histograms need explanation.

Author Response

We are thankful for the helpful suggestions from the Reviewers and Editor.

We have tried our best to revise the manuscript carefully. Here we resubmit a new version of our manuscript, which has been modified according to the Reviewers' comments. And a point-to-point response to reviewers has been given below.

Thank you very much for your patient and intelligent work on our manuscript. Your works are very valuable for us to improve the quality of this paper.

Point 1: In the title, some words are capitalized, and some are small. Please unify them.

Response 1: We appreciate your comment on the title. The first letter case of the title has been revised.

Point 2: Please, check the abbreviations throughout the manuscript. You use AX, AXs, and sometimes AXx.

Response 2: Thanks! AX and AXs are abbreviations of arabinoxylan, and AXX is the arabinoxylan treated with endo-1,4-β-xylanase, which is different from AX. For the convenience of reading, we have unified AX and AXs as AX, and attached the abbreviation word list

Point 3: β symbol should be italic in β-d-xylopyranose, β-(1,4)-glycosidic, and other words.

Response 3: Thanks! The symbols have been modified as you suggested in revised version.

Point 4: Line 108 1H NMR instead of H NMR.

Response 4: Thanks! The manuscript has been modified as you suggested in revised version.

Point 5: The author should give the reference in the Materials & Methods section, Treatment of cells with arabinoxylan hydrolysates and Treatment of cells with MyD88 inhibitor.

Response 5: Thanks! The references have been added in the Materials & Methods section.

Point 6: 1H NMR results need more clarification.

Response 6: Thanks! 1H NMR results have been clarified as you suggested in discussion section.

Point 7: Please, write Figure instead of Fig.

Response 7: Thanks! The words have been modified as you suggested in revised version.

Point 8: The authors should highlight their study. More discussion could be put in this part, and some comparison between this study and other investigations should be mentioned.

Response 8: Thanks! The discussion has been modified as you suggested in revised version.

Point 9: Some grammatical, alignment and typographical errors are noted in the manuscript, and they should be thoroughly checked and corrected throughout the manuscript.

Response 9: Thanks! The manuscript has been modified as you suggested in revised version.

Point 10: Some symbols or letters on histograms need explanation.

Response 10: Thanks! The symbols or letters have been explained in the figure note.

Reviewer 2 Report

This is an in vitro study showing regulatory role of Arabinoxylan derivatives (AXs) on the experimental intestinal barrier function, and the authors found that both molecular weight and degree of substitution of the AXs are related to their functions. The study was carried precisely, and the results were clear. 

Major point to be clarified: Concentration of the AXs used in the experiments and numbers of experiments performed should be described in the legends of Figure 4, 5, 6, and 7.

Minor point: “Occluding” should be corrected as “Occludin” in Figure 5.  

Author Response

We are thankful for the helpful suggestions from the Reviewers and Editor.

We have tried our best to revise the manuscript carefully. Here we resubmit a new version of our manuscript, which has been modified according to the Reviewers' comments. And a point-to-point response to reviewers has been given below.

Thank you very much for your patient and intelligent work on our manuscript. Your works are very valuable for us to improve the quality of this paper.

Point 1: Major point to be clarified: Concentration of the AXs used in the experiments and numbers of experiments performed should be described in the legends of Figure 4, 5, 6, and 7.

Response 1: Thanks! Concentration of the AXs (400 μg/mL) used in the experiments has been added in the Materials & Methods section, and the number of experiments performed (n = 3) has been described in the legends of Figure 4, 5, 6, and 7.

Point 2: Minor point: “Occluding” should be corrected as “Occludin” in Figure 5.

Response 2: Thanks! The figures have been modified as you suggested in revised version.

Reviewer 3 Report

The manuscript entitled “Different structures of arabinType of the Paperoxylan hydrolysates alleviated Caco-2 cell barrier damage by regulating the TLRs/MyD88/NF-κB pathway” and authored by Li et al showed that AXx regulated the expression of inflammatory factors IL-2, TNF-α, IL-6 and IL-10. Authors then showed that AXx reduced the intestinal barrier damage induced via inhibiting the TLRs/MyD88/NF-κB pathway and activating the TLRs/PKC pathway. They concluded that AXs with higher Ds and Mw might be better in alleviating intestinal barrier damage, and MyD88 might be the key point of AXx to identify these signaling pathways. Being a polysaccharide present in the cell wall of cereals like wheat, barley, rice, and others, before focusing on arabinoxylan, more inclusive background should be provided to briefly introduce health-promoting effects of natural products at a wider perspective. Results of the following study (and or others) would be useful and should be integrated: https://www.nature.com/articles/s41598-021-86391-z

One major concern is the uncertainty associated with the statistical analyses in this study.

Other comments

·       Careful proofreading is absolutely a MUST.

·       The statistical significance shown in fig 4 does not seem to support the presented discussion.

·       Uncropped gels should also be added to the supplementary data.

·       What do “a and b” exactly refer to in same figure? Also, quote “---- a-d differ significantly (p < 0.05).”, there are no “c” and “d” in figure 4?!

·       Fig 6A and B should be merged in each assessed marker. Same should be the case for fig 7A and B.

·       

Author Response

We are thankful for the helpful suggestions from the Reviewers and Editor.

We have tried our best to revise the manuscript carefully. Here we resubmit a new version of our manuscript, which has been modified according to the Reviewers' comments. And a point-to-point response to reviewers has been given below.

Thank you very much for your patient and intelligent work on our manuscript. Your works are very valuable for us to improve the quality of this paper.

Point 1: One major concern is the uncertainty associated with the statistical analyses in this study.

Response 1: Thank you so much for your intelligent work to improve the quality of this paper. Your comments are very helpful for us. We have done our best to amend this work. And the manuscript has been carefully revised according to the comments. The following is a point-to-point response.

Point 2: Careful proofreading is absolutely a MUST.

Response 2: Thanks! The manuscript has been revised through careful proofreading.

Point 3: The statistical significance shown in fig 4 does not seem to support the presented discussion.

Response 3: Thanks! Figure 4 has shown a good relationship between the integrity of Caco-2 cells monolayer and TER values. However, there is no significance in these groups, and we have illustrated our results in the results and discussion sections.

Point 4: Uncropped gels should also be added to the supplementary data.

Response 4: The data for uncropped gels was provided in the supplementary information file during submission process, please check it. If not so, we are sorry for the error and we have resubmitted the supplementary data.

Point 5: What do “a and b” exactly refer to in same figure? Also, quote “---- a-d differ significantly (p < 0.05).”, there are no “c” and “d” in figure 4?!

Response 5: Thanks! Values with different superscript letter are significantly different (p <0.05). “a and b” indicate that there is a significant difference between the two sets of data. The letters “c and d” have been deleted in figure 4.

Point 6: Fig 6A and B should be merged in each assessed marker. Same should be the case for fig 7A and B.

Response 6: Thanks! The figures have been merged as you suggested in revised version.

Round 2

Reviewer 1 Report

The authors revised the manuscript according to the recommendations. Only one item should be revised, the headings and subheadings in MDPI, I think they are in sentence case not capitalized each word.

Reviewer 3 Report

none